# Continuous Production of Dietetic Structured Lipids Using Crude Acidic Olive Pomace Oils

**DOI:** 10.3390/molecules28062637

**Published:** 2023-03-14

**Authors:** Joana Souza-Gonçalves, Arsénio Fialho, Cleide M. F. Soares, Natália M. Osório, Suzana Ferreira-Dias

**Affiliations:** 1Instituto Superior de Agronomia, Universidade de Lisboa, LEAF—Linking Landscape, Environment, Agriculture and Food—Research Center, Associated Laboratory TERRA, 1349-017 Lisbon, Portugal; 2Department of Bioengineering, Instituto Superior Técnico, Universidade de Lisboa, 1049-001 Lisbon, Portugal; 3iBB—Institute for Bioengineering and Biosciences and i4HB—Institute for Health and Bioeconomy, Instituto Superior Técnico, 1049-001 Lisbon, Portugal; 4Institute of Technology and Research (ITP), Avenida Murilo Dantas 300—Farolandia, Aracaju 49032-490, Brazil; 5Tiradentes University (UNIT), Avenida Murilo Dantas 300—Farolandia, Aracaju 49032-490, Brazil; 6Instituto Politécnico de Setúbal, Escola Superior de Tecnologia do Barreiro, 2839-001 Lavradio, Portugal; 7Instituto Superior de Agronomia, Universidade de Lisboa, Centro de Estudos Florestais, Associated Laboratory TERRA, 1349-017 Lisbon, Portugal; 8Instituto Superior de Agronomia, Universidade de Lisboa, Laboratório de Estudos Técnicos, 1349-017 Lisbon, Portugal

**Keywords:** acidolysis, biocatalysis, continuous bioreactor, dietetic lipid, interesterification, lipase, low-calorie fat, olive pomace oil, structured lipids, structured triacylglycerols

## Abstract

Crude olive pomace oil (OPO) is a by-product of olive oil extraction. In this study, low-calorie structured triacylglycerols (TAGs) were produced by acidolysis of crude OPO with medium-chain fatty acids (caprylic, C8:0; capric, C10:0) or interesterification with their ethyl ester forms (C8EE, C10EE). These new TAGs present long-chain fatty acids (L) at position *sn*-2 and medium-chain fatty acids (M) at positions *sn*-1,3 (MLM). Crude OPO exhibited a high acidity (12.05–28.75% free fatty acids), and high contents of chlorophylls and oxidation products. Reactions were carried out continuously in a packed-bed bioreactor for 70 h, using *sn*-1,3 regioselective commercial immobilized lipases (*Thermomyces lanuginosus* lipase, Lipozyme TL IM; and *Rhizomucor miehei* lipase, Lipozyme RM IM), in solvent-free media at 40 °C. Lipozyme RM IM presented a higher affinity for C10:0 and C10EE. Lipozyme TL IM preferred C10:0 over C8:0 but C8EE over C10EE. Both biocatalysts showed a high activity and operational stability and were not affected by OPO acidity. The New TAG yields ranged 30–60 and the specific productivity ranged 0.96–1.87 g NewTAG/h.g biocatalyst. Lipozyme RM IM cost is more than seven-fold the Lipozyme TL IM cost. Therefore, using Lipozyme TL IM and crude acidic OPO in a continuous bioreactor will contribute to process sustainability for structured lipid production by lowering the cost of the biocatalyst and avoiding oil refining.

## 1. Introduction

Lipids are considered to be the main source of energy, essential fatty acids, vitamins and antioxidants. Some of them can be classified as functional foods, as is the case of virgin olive oil. Olive oil is recognized as one of the major fat sources in the Mediterranean diet. According to the data from the International Olive Council (IOC), the world production of olive oil reached 3099 kton in the crop year 2021/2022 (provisional values), 64% of it produced in the European Union (1983 ktons), with Portugal being the fourth producer and consumer in Europe [1]. Unexploited agro-industrial residues are a worldwide major concern in terms of environmental problems. Therefore, valorizing the by-products that result from olive processing is important. The olive pomace is obtained after olive oil extraction by mechanical processes. This residue still contains about 3–4.5% (wet basis) of residual oil, known as olive pomace oil, with a composition similar to that of olive oil [2]. From the most recent world statistics on olive oil, ca. 4.716 billion tons of dry olive pomace containing 282.960 kton of crude olive pomace oil are estimated to be obtained every year. 

Oils obtained from agro-food residues and waste (e.g., grapeseed, olive pomace, and spent coffee grounds oils) can be used as raw materials to produce structured lipids (SLs) [3,4,5]. The direct use of crude oils is a viable option because the sustainability of the process will greatly increase since the refining process of the oil is not needed.

SLs are modified lipids (fats and/or oils) that do not exist in nature, presenting improved technological, functional, and/or pharmaceutical properties. SLs are currently defined as triacylglycerols (TAGs) or phospholipids that have been (i) modified by the incorporation of new fatty acids (FAs), (ii) restructured to change the positions of FAs or the fatty acid (FA) profile, from the natural state, or (iii) synthesized to yield novel TAGs (or phospholipids), either chemically or enzymatically [6,7,8]. Functionality, physical properties, metabolic fate, and health benefits associated with lipids depend on FA composition of TAGs and FA distribution on the TAG glycerol backbone. Different types of SLs, namely, modified fat blends to obtain adequate rheological properties for margarine and shortening manufacture, oils enriched in omega-3 polyunsaturated fatty acids (omega-3 PUFA), cocoa butter equivalents, human milk fat substitutes and dietetic SLs, and respective properties have been described in several reviews [6,7,8,9,10,11,12]. 

In the food industry (e.g., margarine and shortenings plants), plastic fats have been obtained by interesterification of fat blends currently carried out using non-regioselective chemical catalysts, acting under reduced pressure, and at temperatures up to 270 °C. The equilibrium is achieved in less than 2 h of reaction [13]. 

The enzymatic interesterification of fat blends has also been performed, batch-wise or in continuous packed-bed or fluidized-bed bioreactors, to obtain modified *trans*-free fat blends for margarine and shortenings, rich in specific fatty acids [14,15,16,17,18,19,20,21]. Xie and Zang performed the enzymatic modification of the following vegetable oils to produce *trans*-free plastic fats: (i) soybean oil by interesterification with lard [17] and with methyl stearate or palm stearin [18], and (ii) rice bran oil, by interesterification with palm stearin [19]. When non-regioselective lipases are used, they act at random in the TAG backbone and the obtained products present a similar composition to those obtained by chemical interesterification [14,17,18,19]. The enzymatic interesterification using non-regioselective biocatalysts is a low energy consumption process because a lower reaction temperature is used. The use of methyl esters [18] as acyl donors should be avoided since methanol will be released in the reaction. Methanol is not allowed in food products and may inactivate the lipase during the reactions [22]. The interesterification of palm olein alone [20] or with lard [21] was carried out in a solvent-free stirred tank or a packed-bed bioreactor, respectively, using immobilized *sn*-1,3 regioselective *Thermomyces lanuginosus* lipase (Lipozyme TL IM) as a catalyst. The use of *sn*-1,3 regioselective lipases will allow the maintenance of long-chain mono- or polyunsaturated fatty acids at position *sn*-2 in TAGs. This is of utmost nutritional importance since long-chain unsaturated fatty acids, namely, the essential fatty acids, are better absorbed by the human body in the form of *sn*-2 monoacylglycerols [23].

Among SLs, dietetic structured lipids, also known as low-calorie TAGs, are a type of lipid containing a specific combination of medium-chain fatty acids (M) at positions *sn*-1 and *sn*-3, and a long-chain fatty acid (L) at position *sn*-2 (MLM). In vegetable oils, mono- and polyunsaturated long-chain fatty acids, namely, essential FAs, are preferentially located at position *sn*-2. Medium-chain fatty acids have lower caloric value than long-chain FAs and are metabolized in the liver like glucose, and not accumulated as fat in the human body. Therefore, MLM SLs will provide the essential FAs for the humans and, at the same time, will contribute to obesity control due to their lower caloric value than that of natural fats (ca 5 kcal/g against 9 kcal/g natural fats) [11,12]. These properties make MLM useful for reducing the possibility of obesity and promoting proper body function. 

There are several lipids in the market, containing medium-chain FAs, for nutritional and pharmaceutical purposes. The company Stepan Lipid Nutrition commercializes medium-chain TAGs as energy source for people with malabsorption syndrome and for several food and pharmaceutical applications, due to their low-calorie content [24]. These products (with the trade name NEOBEE) are obtained by the esterification of glycerol with mixtures of caprylic and capric acids, obtained by fractionation of palm kernel and coconut fats. Similar products (CAPTEX 300 and CAPTEX 355) are commercialized by ABITEC Company [25]. However, these TAG products do not have long-chain FAs, which are very important for human nutrition. To overcome this problem, other TAGs containing medium- and long-chain FAs are already in the market. For example, in 2003, Benefat (or Salatrim: abbreviation of “Short- and long-chain acyl triglyceride molecule”) was approved by the EU as a low-calorie alternative to natural oils and fats. This product, consisting of blends of TAGs containing medium- and long-chain fatty acids, was launched by Danisco and is used in confectionery, bakery, and dairy products [26].

Interesterification and acidolysis are the most common methods used for the production of low-calorie SLs, characterized by the use of ethyl/methyl esters or free fatty acids (FFA) as acyl donors in the reaction, respectively. Since the use of chemical catalysts does not allow a specific action, their replacement by biocatalysts has been increasing. In this sense, the use of *sn*-1, 3 regioselective lipases is mandatory since they cut FAs only at positions *sn*-1 and *sn*-3 of TAGs and produce lipids with unique structures and characteristics. 

The synthesis of SLs by immobilized enzymes in solid carriers can be performed in batch or in continuous bioreactors, either in the presence of an organic solvent or in solvent-free media. At laboratory scale, batch stirred tank reactors (BSTR) have been currently used. BSTR can be a good option if the support particles are not damaged by shear forces. This configuration is adequate to situations of substrate inhibition. However, for process implementation in continuous mode, packed (PBR) or fluidized-bed (FBR) tubular bioreactors are preferred to completely stirred tank reactors. Continuous PBR are the most adequate reactors to repress acyl-migration, due to shorter residence times needed [15,27,28]. Fluidized-bed reactors allow for high mass transfer rates and low shear stress for the immobilized biocatalyst particles [14,29].

Lipozyme TL IM (immobilized lipase from *Thermomyces lanuginosus*) and Lipozyme RM IM (immobilized lipase from *Rhizomucor miehei*) are examples of commercial enzymes immobilized on silica gel and microporous anion exchange resin, respectively. Given the lack of mechanical resistance to stirring usually applied to batch mode, easy desorption, or enzyme leakage from the support during batch reuse, Lipozyme TL IM is not adequate for use in this type of bioreactor [30]. Apart from the reuse in batch mode, this type of bioreactor has lower costs, although it is difficult to control heat transfer. In continuous bioreactors, it is possible to decrease the costs associated with enzymatic processes.

Several oils rich in long-chain FAs have been used to produce low-calorie structured lipids either by acidolysis or interesterification reaction, in batch stirred bioreactors (e.g., olive, olive pomace, linseed, spent coffee grains, grapeseed, avocado, or microbial oils), catalyzed by *sn*-1,3-regioselective lipases [3,4,5,31,32,33,34,35,36]. A batch packed-bed reactor of Lipozyme TL IM, working under continuous recirculation mode, was successfully used to obtain oil blends rich in medium- and long-chain TAGs by interesterification of soybean oil with TAGs rich in caprylic and capric acids [37]. The implementation of enzymatic synthesis of MLM in continuous PBR with commercial immobilized *Rizomucor miehei* lipase (Lipozyme IM) was carried out by Xu et al. [28]. In this study, SLs were obtained by the (i) acidolysis of refined rapeseed oil with capric acid or safflower oil with caprylic acid or (ii) interesterification between medium-chain TAGs and oleic acid, in a solvent-free system. In continuous mode, the biocatalyst maintained 60% of the initial activity after a 4-week continuous operation. Lipozyme TL IM was also used in a continuous PBR for the interesterification between medium-chain TAGs and fish oil, in the absence of organic solvent [28]. In this system, the activity of Lipozyme TL IM was maintained along a 2-week operation. 

Previous studies were performed by our group to valorize crude OPO as feedstock for the production of SLs, by batch interesterification of crude acidic olive pomace oils with caprylic and capric acids, or their ethyl esters, in solvent-free media. In these studies, *Rhizopus oryzae* lipase immobilized in magnetic nanoparticles or commercial immobilized lipases (Lipozyme RM IM and Lipozyme TL IM) were used as biocatalysts [4,38]. However, according to our knowledge, the production of these SLs with crude olive pomace oil, in continuous bioreactors, has not been studied yet. Therefore, the aim of the present study was to implement the lipase-catalyzed production of structured lipids with low calorie value, in continuous packed-bed bioreactor, with known advantages in terms of process scale-up, using crude OPO. The direct use of crude high acidic oil, instead of refined oil, will be an important contribution to decrease production costs and make the process economically feasible. Moreover, due to the low mechanical resistance of Lipozyme TL IM, its use in a continuous packed-bed reactor may be an option to preserve biocatalyst integrity. Acidolysis reactions with capric and caprylic acid or interesterification with the respective ethyl esters were performed. Studies related to specific productivities and costs based on experimental data can promote the possibility of developing scale-up activities and their application in bioindustries. 

## 2. Results and Discussion

### 2.1. Crude Olive Pomace Oil Characterization

All crude olive pomace oil (OPO) samples were centrifuged to remove most of the impurities prior to analysis and their use as a reaction substrate. Table 1 shows the chemical characterization of OPO samples, namely, acidity, UV absorbance (K_232_ and K_270_), chlorophyll pigments, and fatty acid composition. 

In terms of oil quality, the acidity consists of the amount of FFA released by the hydrolysis of TAGs. All crude oil samples presented high acidity: 12.05% (OPO-1), 15.06% (OPO-2), and 28.75% (OPO-3). The oxidation products are also related with the quality of oils. Each absorbance at 232 nm and 270 nm are provided by the specific extinction coefficient (K_232_: indicator of the presence of initial products of oxidation; K_270_: indicator of the presence of final oxidation products) [39]. The results for all OPO are considered high (about K = 5), which suggests a high oxidation stage. 

The increase in acidity and K values, from OPO-1 to OPO-3, may be related to the storage conditions of olive pomace before oil extraction, promoting hydrolysis and oxidation processes of TAGs. In fact, in this study, the OPO oils were solvent-extracted from olive pomaces obtained in two-phase continuous olive oil decanters. These pomaces are very rich in water (ca. 70%). Along olive oil extraction (Autumn/Winter; October–December in the northern hemisphere), the pomaces are currently discarded in large open-air ponds, where they can stay for months until the oil is extracted. These conditions promote the hydrolysis and oxidation of the oil, among other degradation reactions. The presence of oxidation products in the oils has been related to an inhibitory effect on lipase activity [15].

Chlorophyll pigments are responsible for the green color intensity of olives. There is no defined limit for the measurement of chlorophyll pigments in OPO. Chlorophyll pigment values varied from 367.0 to 447.5 mg of pheophytin a per kg of OPO. The high content of chlorophyll pigments in OPO, compared with the content in virgin olive oils, are a result of solvent extraction of crude oils. These pigments have pro-oxidant activity, which means they will catalyze the oxidation reaction, contributing for the quality decrease of the oils [40]. Moreover, OPO-2 shows higher contents of chlorophyll pigments than OPO-1 and OPO-3. The differences might be due to changes in olive cultivars and stages of fruit development [41]. 

Regarding the fatty acid (FA) composition, all OPO samples had a similar composition which was similar to FA composition of olive oil, as reported by the Commission Regulation (EEC) Nr. 2568/91 [39] for both oils: oleic acid (C18:1) is present in the largest amount (69–71%), followed by palmitic acid (C16:0) (14%), and linoleic acid (C18:2) (10–11%). Stearic acid (C18:0) accounted for 2.11–2.23%, whereas palmitoleic acid (C16:1) contributed to 1.32–1.46% (Table 1).

It shows that the OPO has good nutritional quality. Moreover, it is a good source of oleic and linoleic acids, mainly esterified at position *sn*-2 in TAGs, which is important in the prevention of cardiovascular diseases.

### 2.2. SL Synthesis

OPO acidolysis and interesterification were carried out in solvent-free media, catalyzed by Lipozyme TL IM or Lipozyme RM IM, in a packed-bed continuous bioreactor, at 40 °C. The acidolysis reactions of OPO were developed with C8:0 or C10:0, while in interesterification, C8EE and C10EE were used as acyl donors. Three different OPO samples were used to perform the reactions. OPO-1 (12.05% FFA) was used in the experiments with C10EE using Lipozyme TL IM or Lipozyme RM IM, and in the acidolysis with C10:0 using Lipozyme TL IM. OPO-2 (15.06% FFA) was used in the acidolysis with C10:0 catalyzed by Lipozyme RM IM. OPO-3 (28.75% FFA) was employed in interesterification with C8EE and acidolysis with C8:0, either with Lipozyme TL IM or Lipozyme RM IM.

In both types of reaction, the original FA at positions *sn*-1 and *sn*-3 of the TAG molecules will be replaced by caprylic or capric acid, in the presence of a *sn*-1,3 regioselective lipase, with the formation of novel TAGs. The new TAG yield, TAG conversion and FFA (or FA ethyl esters) conversion were determined. Figure 1 and Figure 2 show TAG conversion and New TAG yield along 70 h continuous acidolysis or interesterification reactions catalyzed by Lipozyme TL IM (Figure 1) and Lipozyme RM IM (Figure 2). In continuous mode, the reactions can be considered as steady state. However, in two reactions (C8EE + Lipozyme TL IM and C10EE + Lipozyme RM IM), the time to attain the steady state took longer to achieve. This phenomenon might be due to a lack of uniformity in the enzyme bed, making mass transfer more difficult (Figure 1 and Figure 2) [42].

Both biocatalysts presented a high activity along all the continuous 70 h reactions. The behavior of the biocatalysts along the reactions seemed not to be affected by the acidity value of OPO used which varied from ca. 12 to 29% (Table 1). Similar behavior was previously observed for Lipozyme RM IM in batch acidolysis of crude pomace oils with acidity ranging from 3.4 to 20% and caprylic or capric acids [38]. It is worth mentioning that the acidity of OPO is due to long-chain FFA in the reaction media, mainly oleic acid (Table 1). 

Conversion values of TAGs were approximately constant along the 70 h operation of the bioreactor. The values were greater than 70% when using both lipases, either in acidolysis or interesterification (Figure 1 and Figure 2). Values of around 80% TAG conversion were attained in acidolysis with capric acid or caprylic acid, catalyzed by Lipozyme RM IM, and in all interesterification systems. 

The novel TAG molecules generated might be of MLL or MLM type, depending on the FA substitution achieved with the acyl donor involved (one or two substitutions of a long-chain FA by caprylic or capric acid, at positions *sn*-1,3, respectively) [8]. Due to acyl migration, the formation of TAGs containing caprylic or capric acid at position *sn*-2 (MML, LML or MMM) may also occur, but in a lesser extent [20]. Since these reactions create intermediate molecules (monoacylglycerols, MAGs, and diacylglycerols, DAGs), the conversion values of TAGs are always higher than the yield of new TAGs along the reactions.

With respect to the initial yields of new TAGs, values around 50–60% were observed for the interesterification reactions with C8EE or C10EE and acidolysis with capric acid catalyzed by both biocatalysts. When caprylic acid was used, new TAG yields were only around 30% (Figure 1 and Figure 2). As expected, new TAG yields were lower than TAG conversion values. The differences between TAG conversion and yield values were about 25–30% for all systems, except for the acidolysis with caprylic acid (C8:0). Either with Lipozyme TL IM or Lipozyme RM IM, this difference was about 40%. This shows that ca. 40% of the original TAGs were converted into partial acylglycerols and not into New TAGs of MLM or MLL type, when C8:0 was used as the acyl donor. This behavior shows the lower affinity of both enzymes for C8:0. Moreover, the new TAG yield observed in the system with C8EE and Lipozyme RM IM decreased to 25% after 70 h continuous operation of the bioreactor (Figure 2).

The observed behavior for both biocatalysts is characteristic of lipases which present a higher affinity for longer-chain fatty acids than for medium- or short-chain fatty acids, conversely to esterases [43]. Therefore, a higher activity is expected with increasing fatty acid chain-length.

The observed results can be also explained by the concept of Log P (Hansch parameter) applied to the acyl donors used in each reaction system. Laane et al. [44] proposed for the first time, the use of Log P to assess the biocompatibility of organic solvents: solvents with Log P lower than 2 are harmful for the biocatalysts because they are very hydrophilic and remove the essential water layer of the biocatalysts; between 2 and 4, their behavior is unpredictable, and values higher than 4 indicate that no inactivation of the biocatalyst will occur. From data published by PubChem [45], the Log P values are as follows: Log P C8:0 = 3.05; Log P C10:0 = 4.09; Log P C8EE = 3.842; and Log P C10EE = 4.861. 

Both capric acid and ethyl decanoate have Log P values higher than 4. When Lipozyme RM IM was used, the highest productivities in SLs were obtained in the presence of these acyl donors. Lipozyme TL IM seemed not to be so much affected with more hydrophilic molecules since it showed similar results also in the presence of C8EE, with a Log P value slightly lower than 4.0. Either for Lipozyme TL IM or for Lipozyme RM IM, the lowest productivities in SLs were obtained in the presence of caprylic acid, which has a Log P value of 3.05. Thus, a possible inactivation of the biocatalysts caused by the more hydrophilic acyl donors may explain the observed results.

In batch bioreactors, Heinzl et al. [38] observed that Lipozyme RM IM achieved similar new TAG yields (47.8–53.4%), either in the acidolysis of crude olive pomace oil with C8:0 or C10:0 or in the interesterification with the respective ethyl esters. Conversely, Lipozyme TL IM showed a better performance in batch interesterification than in acidolysis, and the reaction rates were always lower than with Lipozyme RM IM. Moreover, Lipozyme TL IM showed higher affinity for C10EE over C8EE [38]. 

When crude spent coffee ground oil was used, both Lipozyme RM IM and Lipozyme TL IM showed a higher affinity for C10EE over C8EE, in batch interesterification [5]. However, Lipozyme RM IM produced higher yields in new TAGs by acidolysis than by interesterification. As observed with crude OPO, Lipozyme TL IM showed a higher activity in interesterification than in acidolysis. 

Other oils have been used as feedstock in SL production, with reactions catalyzed by Lipozyme RM IM or Lipozyme TL IM, such as grape seed oil [3] (34.53% C10:0 incorporation) or roasted crude sesame oil, rich in oleic acid [46] (C8:0 incorporation of 42.5%). In several studies, Lipozyme RM IM exhibited a higher degree of incorporation than Lipozyme TL IM [38,47,48,49]. 

In the present study, Lipozyme TL IM in continuous PBR showed a better performance than previously in batch mode. This may be explained by (i) the low mechanical resistance of this biocatalyst to magnetic stirring used in batch bioreactors, which might be responsible for the destruction of some particles promoting enzyme inactivation in contact with the impurities of crude OPO, and/or (ii) by a high enzyme load in the bioreactor bed, which will compensate the slower reaction rates of this biocatalyst compared to Lipozyme RM IM.

### 2.3. Operational Stability of Biocatalysts

The results concerning TAG conversion and new TAG yields over 70 h continuous PBR operation (Figure 1 and Figure 2) show that for the majority of the systems, the values were almost constant during this time. 

Deactivation was only observed for Lipozyme TL IM in the presence of C10:0, and for Lipozyme RM IM in the interesterification with C8EE (Figure 3; Table 2). The observed New TAG yield was considered as 100% activity, when the steady state was reached. As the reaction progressed, the residual activity of the biocatalyst at time t was calculated as the ratio between the observed yield at time t, and the initial yield (100% activity).

The following models, where Act_t_ is the residual activity at time t, were fitted: 

Linear deactivation model (Lipozyme TL IM + C10:0)
(1)$${\mathrm{Act}}_{t}=-0.224t+101.0$$

First-order deactivation model (Lipozyme RM IM + C8EE)
(2)$${\mathrm{Act}}_{t}=95.41e^{\left(-0.01t\right)}\;$$

These model Equations (1) and (2) were used to estimate the half-life time for each biocatalyst in these systems [42]. Thus, the half-life of Lipozyme TL IM in the presence of capric acid and OPO with 12% acidity was estimated as 228 h. For Lipozyme RM IM, in the interesterification of ethyl caprylate with OPO containing 28.75% acidity, the half-life time was lower and equal to 74 h, and the deactivation coefficient was 0.01 h^−1^. 

Table 2 shows that the operational stability of both biocatalysts seems not be affected by the acidity of OPO used.

Most of the studies reported in the literature used batch reactors to conduct MLM structured lipids production. Consistent with our findings in a continuous bioreactor, high operational stability has been reported for lipases used in batch mode, even in the presence of olive pomace oil with high acidity [38]. López-Fernández et al. [50] observed that long-chain free fatty acids had a positive effect on the stability of a recombinant *Rhizopus oryzae* lipase immobilized on a resin, when used in batch transesterification of crude acidic olive pomace oil (19% acidity) or other crude oils with ethanol or methanol, for biofuel production. They concluded that the operational stability of the biocatalyst depended more on the oil type utilized than on its acidity.

Studies on batch acidolysis of coffee ground crude oil with caprylic or capric acids have also shown lower half-lives for Lipozyme RM IM (47 h and 54 h, respectively), compared to our work [4]. In Nunes et al. [31] and Mota et al. [4], a loss of activity of the biocatalyst Lipozyme TM IM was verified in the acidolysis of virgin olive oil or OPO, respectively.

Previous studies in continuous mode, with the same type of bioreactor (PBR), have been used to produce SLs. Lipozyme TL IM was used in a continuous PBR to produce low-calorie SLs by interesterification of refined fish oil with medium-chain TAG blend rich in caprylic and capric acids, at 60 °C, in solvent-free system [28]. This immobilized lipase was stable during 2-week continuous operation. 

During the acidolysis of grapeseed oil with C10:0, using Lipozyme RM IM in a continuous PBR, a half-life of 209.6 h and a deactivation coefficient of 0.0061 h^-1^ were estimated [51]. These results are similar to our results with the same acyl donor for Lipozyme TL IM (228.3 h), but our results were superior when Lipozyme RM IM was used in the same system.

In the acidolysis of roasted sesame oil and caprylic acid in a continuous PBR, Lipozyme RM IM showed a high stability with a half-life of 19.2 days [46].

Lipozyme TL IM and Lipozyme RM IM were also used in the interesterification of blends of olive oil, palm stearin, and palm kernel oil implemented in a continuous PBR [16]. Both biocatalysts followed a first-order deactivation profile showing half-lives of 88 h and 60 h, for Lipozyme TL IM and Lipozyme RM IM, respectively. 

These results strongly suggest that the operational stability of immobilized enzymes relies on the type of bioreactor operation (batch or continuous), on the reaction and on the reaction medium composition. In batch mode, mechanical agitation comes into direct contact with the enzyme. In continuous mode, the enzyme is placed in the packed-bed column without agitation, or in a fluidized-bed, helping the integrity of the enzyme. In this sense, further research into the continuous mode, using OPO as a raw material and immobilized lipases, is required.

The results of the present study are very promising, with regard to the advantage of using continuous bioreactors. Many variables, such as reagent molar ratio, substrate composition, and bioreactor operating parameters have an impact on product yield and quality [51].

### 2.4. Productivity and Biocatalyst Costs 

To evaluate the economic feasibility of the enzymatic reactions, the cost associated with each biocatalyst used was determined. The price for each biocatalyst was provided by Novozymes A/S: Lipozyme TL IM—110 EUR /Kg; Lipozyme RM IM—923 EUR /Kg (information given in September 2022). 

The specific productivity of New TAGs (g/h.g biocatalyst) was calculated using the measured flow rate (Q), the operation time (70 h), the average production of new TAGs during the operation, and the mass of enzyme bed (10 g).

Table 3 shows the specific productivities for the synthesis of novel TAGs in each system and the biocatalyst cost to obtain 1 kg of structured lipids. 

Comparing specific productivities, similar values were obtained with both biocatalysts in the presence of caprylic acid or capric acid and C10EE, being higher when the acidolysis was carried out with capric acid or C10EE. Lipozyme RM IM showed a clear affinity towards capric acid or its ethyl ester. Lipozyme TL IM showed a higher affinity to capric acid over caprylic acid but promoted higher productivities when ethyl caprylate (C8EE) was used instead of ethyl caprate (C10EE).

The cost of Lipozyme TL IM was always lower than EUR 1 per kg of new TAGs, except for the system with caprylic acid, where the specific productivity was almost half the values observed for the other acyl donors. With respect to the cost of Lipozyme RM IM, it varied between EUR 7.06 and 12.73 to produce 1 kg of structured lipids. The highest costs were observed for the interesterification of OPO with C8:0 (EUR 12.73) and C8EE (EUR 11.68), catalyzed by Lipozyme RM IM, where a low yield in New TAGs and considerable deactivation were observed, respectively (Figure 2 and Figure 3, Table 2)

When biocatalyst costs are compared, the use of Lipozyme TL IM is preferred, since Lipozyme RM IM cost varies from 7.3- to 13.4-fold the value for Lipozyme TM IM. Having a similar catalytic behavior and operational stability to Lipozyme RM IM, Lipozyme TL IM showed to be the preferred biocatalyst for the continuous production of dietetic SL from acidic OPO, in a PBR and solvent-free medium, due to its lower price. 

## 3. Materials and Methods

### 3.1. Materials

Crude OPO samples were kindly donated by UCASUL—União de Cooperativas Agrícolas do Sul, Alvito, Beja, Portugal. The commercial immobilized *sn*-1,3 regioselective lipases Lipozyme TL IM (*Thermomyces lanuginosus* lipase immobilized on a non-compressible silica gel carrier; density = 0.40; 250 IUN/g) and Lipozyme RM IM (*Rhizomucor miehei* lipase immobilized on a resin carrier; density = 0.33; 275 IUN/g) were a gift from Novozymes A/S, Denmark. Caprylic acid (C8:0; >98% purity), capric acid (C10:0; >98% purity), ethyl caprylate (C8EE; >98% purity), and ethyl caprate (C10EE; >98% purity) were provided by TCI Europe N.V., Belgium. All reagents used were analytical grade.

### 3.2. Olive Pomace Oil Characterization

To remove the majority of impurities, the OPO was firstly centrifuged (10,000 rpm/30 min, at 40 °C). Three different crude OPO samples were characterized with respect to the following properties: acidity, oxidation products, and chlorophyll pigments and fatty acid (FA) composition (Table 1). 

Both oil acidity and oxidation products were determined in accordance with Commission Regulation EEC, No 2568/91 (1991), in relation to the properties of olive and olive pomace oils and associated analysis procedures [39]. The acidity of the samples was measured based on the amount of free fatty acids (FFA) present in OPO, expressed as oleic acid. Absorbances at 232 and 270 nm were used to analyze oxidation products (K_232_: presence of initial products of oxidation; K_270_: presence of final oxidation products). Both were determined using a UV/Vis spectrophotometer (Agilent Technologies Cary series 100 UV-Vis). The color characterization was determined according to Pokorny et al. [40] by measuring the absorbance at 630, 670, and 710 nm against air to quantify chlorophyll pigments on the samples. Green pigment contents were expressed as pheophytin a (mg/kg). 

Furthermore, the fatty acid composition was evaluated by gas chromatography, after their conversion into fatty acid methyl esters (FAME) using a Perkin Elmer Autosystem 9000 gas chromatograph (GC), equipped with a FID and a fused silica capillary column Supelco SPTM—2380 (60 m × 0.25 mm × 0.2 μm film thickness). The injector and FID were set at 250 and 260 °C, respectively. The column temperature was set at 165 °C for 45 min, increasing at a rate of 7.5 °C/min up to 230 °C, holding for 25 min at this temperature. Helium was used as carrier at a pressure of 20.0 Psig. FAME standards (GLC-10 FAME mix, 1891-1AMP, from Sigma-Aldrich, St. Louis, MA, USA), analyzed under the same conditions, were used for fatty acid identification.

### 3.3. Continuous Bioreactor Systems for Acidolysis and Interesterification Reactions

Interesterification and acidolysis reactions were carried out in solvent-free media, using a continuous packed-bed bioreactor. It consisted of a glass column with an internal diameter of 2 cm and a height of 20 cm, with a double jacket for water circulation and temperature maintenance, and a glass sieve (G0) on the bottom to retain the biocatalyst. For each experiment, 10 g of immobilized lipase was used at its original water activity: 0.152 for Lipozyme TL IM and 0.585 for Lipozyme RM IM, at 20 °C. Reaction media were pumped upwards with a peristaltic pump, to avoid bed clogging and channeling, through silicone tubing from a conical flask placed in a water bath at 40 °C, under magnetic stirring. 

Reaction media consisted of blends of crude OPO with capric acid ethyl ester (C10EE) or caprylic acid ethyl ester (C8EE) in interesterification reactions, or with the acid forms (C10:0 and C8:0, respectively) in the acidolysis reactions. A 1:2 molar ratio of oil/acyl donor, corresponding to the stoichiometric ratio, was used. The bioreactor operated for 70 h, with a pause during the night. Periodically, along the continuous reactions, 5 mL samples were taken. The collected samples were then frozen at −18 °C for subsequent analysis.

For each packed-bed reactor system, the flow rate (Q) was calculated as the ratio of the recovered volume (V, mL) of reaction media to the time (t, min). The residence time (τ) for each system was estimated using the following equation (Equation (3)), described as the ratio between the product of the volume of the enzyme bed (V_enzyme bed_) by its porosity (ε) and the flow rate [27].
(3)$$\tau =\frac{V_{\mathrm{enzyme}\;\mathrm{bed}}*\varepsilon }{Q}$$

Table 4 shows the flow rate (Q, mL/min), enzyme bed volume (V, mL), void fraction (ε), and residence time (τ, min) values for each continuous packed-bed bioreactor working with Lipozyme TL IM or Lipozyme RM IM.

The bed volume with 10 g Lipozyme TL IM was 19.2 mL, while with 10 g Lipozyme RM IM, it was 30.8 mL. The observed flow rate in both cases was 0.6 mL/min. The void fraction was estimated by volume displacement, according to the methodology presented by Xu et al. [27]. For Lipozyme TL IM, the void fraction was 0.34 and the residence time was 11 min. The void fraction for Lipozyme RM IM was 0.39, and the predicted residence time was 20 min, which is about twice the value for Lipozyme TL IM. 

The differences in residence time may be explained by differences in column packaging originating different bed volumes. Xu et al. [27] obtained different values of void fraction in a packed-bed column filled twice with Lipozyme RM IM (0.44 ± 0.01 and 0.47 ± 0.02). For Lipozyme TL IM in a packed-bed reactor, a void fraction of 0.51 ± 0.1 was estimated by Xu et al. [28]. 

The starting operation time was considered when the steady state was attained, i.e., the time equal to four residence times (around 44 min and 80 min when Lipozyme TL IM or Lipozyme RM IM were used, respectively).

### 3.4. Reaction Product Analysis by Gas Chromatography

The different compounds present in the samples, namely the original TAGs and FFA or ethyl esters, novel TAGs, diacylglycerols (DAGs), monoacylglycerols (MAGs) and FFA, were identified and quantified by GC in accordance with European Standard EN 14105 [52] with some modifications. Before GC analysis, all samples needed to be derivatized, following the following steps: (i) 0.05 g of each sample was weighed in a 10 mL volumetric flask and then filled with hexane; (ii) 0.5 mL of the previous mixture was added to a pear-shaped flask; (iii) the solvent was evaporated by rotavapor equipment (<120 mbar, 30–35 °C, 10 min); (iv) 400 μL of internal standard (IS) solution (0.05 g of monononadecanoine in 25 mL of tetrahydrofuran), 200 μL of pyridine, and 200 μL of N-methyl-N-trimethyl-silyl-triflouracetamide were added to the dried sample. After 15 min, (v) 4 mL of heptane was added to the flask under stirring. The sample was transferred to 1 mL vials for gas chromatography analysis. A gas chromatograph Agilent Technologies 7820 A, equipped with an on-column injector, one Agilent J&W DB5-HT capillary column (15 m × 0.320 mm ID × 0.10 μm film), and a flame ionization detector was used.

Helium gas was used as a carrier. The gases were delivered at a flow rate of 30 mL/min (helium, hydrogen, and nitrogen) and 300 mL/min for compressed air. 

The following oven temperature program was followed: an initial temperature of 50 °C rising at a rate of 15 °C/min to 180 °C. Following this, the temperature increased at a rate of 7 °C/min until 230 °C. Finally, the temperature increased to 365 °C over 12 min at a final rate of 10 °C/min. 

Each peak in the chromatograms was identified by comparing it to pure compound standards and with the chromatograms presented in the European Standard EN 14105 [52]. 

Figure 4 shows an example of two chromatograms obtained for the initial reaction medium (OPO and C8EE) and at a steady state of continuous interesterification, catalyzed by Lipozyme TL IM.

In each chromatogram, the area of each peak was divided by the area of the peak of the internal standard (IS). The amounts of initial TAGs were proportional to the sum of the areas of all TAG peaks (A_initial TAG_) divided by the area of the peak of the IS (A_IS_), in the initial fat blend. Along bioreactor operation, the same procedure was used to quantify the new TAGs that appear as new peaks in the TAG region of the chromatogram (A_new TAG_) (Figure 4). Therefore, the New TAG yield (Y,%) was calculated as follows (Equation (4)), where the numerator corresponds to the peaks of the chromatogram at time t, and the denominator corresponds to the peaks of the chromatogram of the initial blend:(4)Y=(∑Anew TAGAIS∑ApeaksAIS)(∑Ainitial TAGAIS∑ApeaksAIS)×100

TAG, FFA, or ethyl esters conversions were calculated as the ratio between the amount of TAG (ethyl ester or FFA) consumed and the corresponding initial quantities (in terms of peak areas divided by the area of the IS peak, as explained for New TAG yield).

### 3.5. Assessment of Continuous Operational Stability of Biocatalysts

Along the continuous bioreactor operation, the activity of the biocatalysts was evaluated as the yield of new TAGs, formed by the incorporation of medium-chain FAs in the TAGs of OPO, either by acidolysis with caprylic or capric acids, or interesterification with their ethyl esters. The residual activity along continuous operation was determined as the ratio between New TAG yield at time t and the New TAG yield, observed at time 0, i.e., when the steady state was attained. The activity at time zero was considered as 100%. When biocatalyst deactivation was observed along the operation, deactivation kinetic models were fitted by the “Solver” add-in in Excel to estimate the operational half-life time (operation time needed to reduce the original activity to 50%) [38].

## 4. Conclusions

This is the first work on the lipase-catalyzed continuous production of low-calorie SLs in a packed-bed reactor, using crude acidic (12–29% acidity) OPO in solvent-free media. Both immobilized lipases used (Lipozyme TL IM and Lipozyme RM IM) presented a high activity either in acidolysis with caprylic or capric acid or in interesterification with their ethyl esters. The SLs were new TAGs where a medium-chain fatty acid was esterified at one of the positions *sn*-1 or *sn*-3 or in both (MLL or MLM molecules).

Yields of new TAGs of around 50–60% were observed for the interesterification reactions with C8EE or C10EE and acidolysis with capric acid, catalyzed by both biocatalysts. In the acidolysis with caprylic acid, new TAG yields were only around 30%. 

The biocatalysts maintained the activity along the 70 h continuous reactions except Lipozyme TL IM in the presence of capric acid (half-life time of 228 h) and Lipozyme RM IM in the interesterification with C8EE (half-life time of 74 h). The acidity value of OPO used (12 to 29% free fatty acids) did not affect the activity and stability of the biocatalysts.

Comparing specific productivities in new TAGs, similar values were obtained in the presence of caprylic acid (0.96 vs. 1.04 g/h.g biocatalyst), capric acid (1.65 vs. 1.79 g/h.g biocatalyst), or C10EE (1.62 vs. 1.87 g/h.g biocatalyst), with Lipozyme TL IM or Lipozyme RM IM, respectively. Higher affinity for C8EE was observed for Lipozyme TL IM, when compared with Lipozyme RM IM, resulting in specific productivities of 1.80 and 1.13 g of new TAGs/h.g biocatalyst, respectively.

Lipozyme TL IM, which is very prone to mechanical damage in batch stirred reactors, maintained its activity over 70 h operation in a packed-bed reactor. Therefore, this biocatalyst is adequate for continuous processes. Moreover, due to its high productivity, stability, and lower cost, Lipozyme TL IM demonstrated to be more promising than Lipozyme RM IM for MLM synthesis in continuous bioreactors. 

Thus, using a crude oil that is a by-product of the olive oil industry, instead of refined oils currently used for SLs production, will contribute to the decrease of the process cost related to oil refining. In addition, the direct use of crude OPO increases process sustainability and reduces the environmental impact, promoting circular economy in the olive oil sector. A longer operation time in continuous bioreactors, as well as new bioreactor configurations, such as fluidized-bed reactors, together with the use of non-commercial biocatalysts, should be tested in further studies to decrease operation costs.

## Figures and Tables

**Figure 1 molecules-28-02637-f001:**
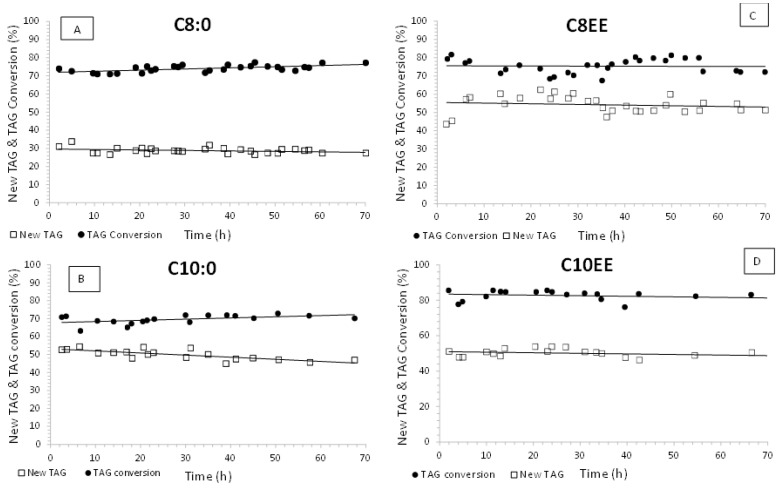
New TAG yield and TAG conversion along continuous acidolysis and interesterification reactions of (**A**) C8:0, (**B**) C10:0, (**C**) C8EE, and (**D**) C10EE, respectively, catalyzed by Lipozyme TL IM biocatalyst.

**Figure 2 molecules-28-02637-f002:**
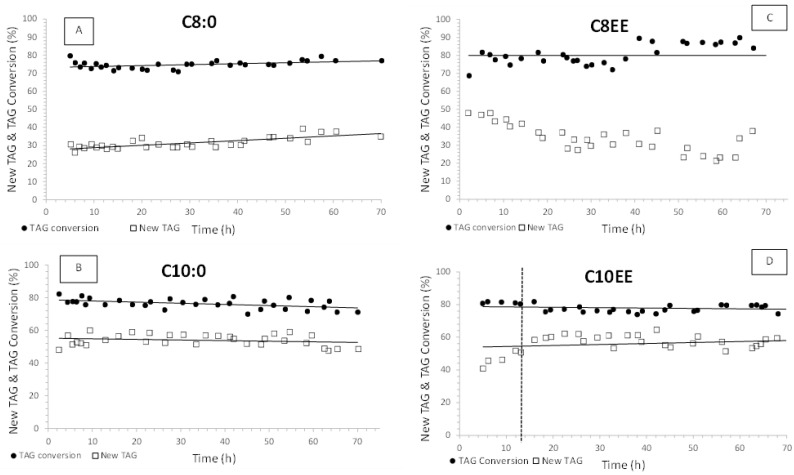
New TAG yield and TAG conversion along continuous acidolysis and interesterification reactions of (**A**) C8:0, (**B**) C10:0, (**C**) C8EE, and (**D**) C10EE, respectively, catalyzed by Lipozyme RM IM biocatalyst.

**Figure 3 molecules-28-02637-f003:**
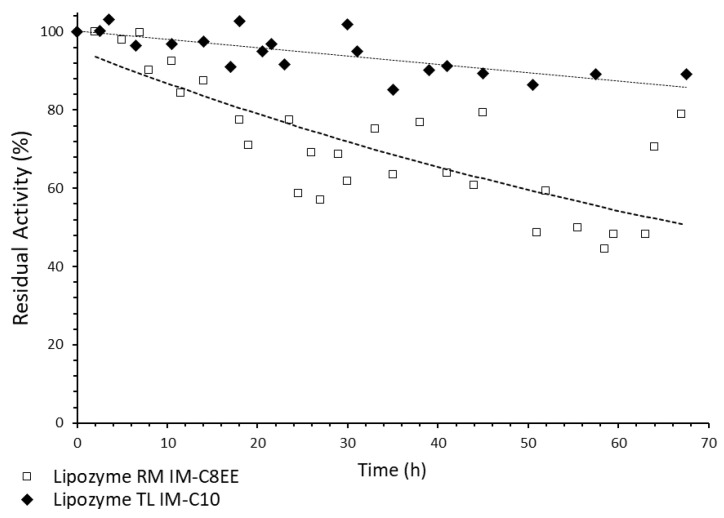
Residual activity assayed by New TAG yield for the acidolysis of OPO with capric acid, catalyzed by Lipozyme TL IM and interesterification of OPO with ethyl caprylate, catalyzed by Lipozyme RM IM, in continuous PBR, and respective deactivation models.

**Figure 4 molecules-28-02637-f004:**
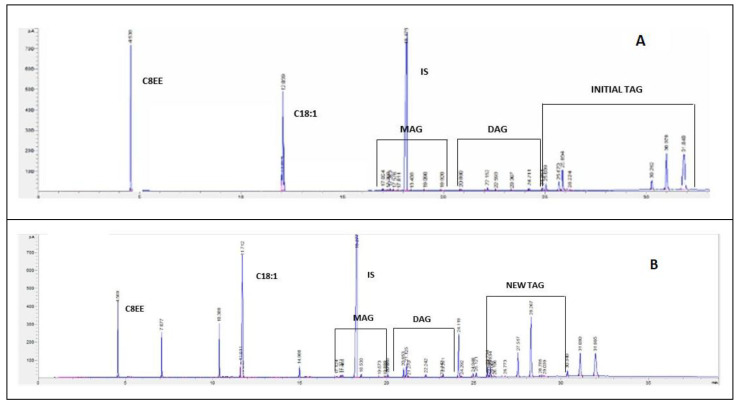
Example of chromatograms obtained (**A**) for the initial reaction medium (OPO and C8EE) and (**B**) at a steady state of continuous interesterification catalyzed by Lipozyme TL IM (C 18:1—peak of oleic acid; IS—peak of the internal standard; MAG—peaks of monoacylglycerols; DAG—peaks of diacylglycerols; Initial and New TAG- peaks of initial TAGs and new TAGs formed during the reaction).

**Table 1 molecules-28-02637-t001:** Chemical characterization of the crude OPO used (OPO-1, OPO-2, and OPO-3): acidity (% free fatty acids), oxidation products (K_232_ and K_270_), chlorophyll pigments, and fatty acid composition (the average value of three replicates ± standard deviation for each sample is indicated).

Characterization Assays	OPO-1	OPO-2	OPO-3
Acidity (%)	12.05 ± 0.45	15.06 ± 0.93	28.75 ± 0.91
K_232_	4.92 ± 0.09	6.18 ± 0.36	6.06 ± 0.71
K_270_	1.90 ± 0.08	2.36 ± 0.16	2.28 ± 0.26
Chlorophyll Pigments (Pheophytin a; mg/kg)	366.97 ± 31.35	447.50 ± 15.84	374.35 ± 11.70
Main FA content (%)			
C18:1	69.05 ± 3.31	70.98 ± 3.41	69.91 ± 3.36
C16:0	14.37 ± 0.69	13.50 ± 0.65	13.45 ± 0.65
C18:2	10.78 ± 1.72	9.64 ± 1.54	10.60 ± 1.70
C18:0	2.11 ± 0.30	2.23 ±0.31	2.35 ± 0.33
C14:0	0.03 ± 0.01	0.03 ± 0.01	0.04 ± 0.01
C16:1	1.46 ± 0.20	1.35 ± 0.19	1.32 ± 0.18
C17:0	0.12 ± 0.02	0.12 ± 0.02	0.11 ± 0.02
C17:1	0.22 ± 0.04	0.22 ± 0.04	0.20 ± 0.04
C18:3	0.84 ± 0.09	0.87 ± 0.10	0.90 ± 0.10
C20:0	0.43 ± 0.05	0.44 ± 0.04	0.47 ± 0.06
C20:1	0.31 ± 0.04	0.31 ± 0.04	0.31 ± 0.04

**Table 2 molecules-28-02637-t002:** Operational stability results for Lipozyme TL IM and Lipozyme RM IM, in a continuous bioreactor (deactivation kinetics model and half-life, h) for acidolysis and interesterification reactions, with OPOs of different acidity values (n.d.: not determined/detected).

Type of Reaction	FFA/Ethyl Esters	Biocatalyst	Deactivation Model	Half-Life (h)	Oil Acidity (%)
Acidolysis	C8:0	Lipozyme TL IM	n.d.	n.d.	28.75
	C10:0		Linear	228	12.05
	C8:0	Lipozyme RM IM	n.d.	n.d.	28.75
	C10:0		n.d.	n.d.	15.06
Interesterification	C8EE	Lipozyme TL IM	n.d.	n.d.	28.75
	C10EE		n.d.	n.d.	12.05
	C8EE	Lipozyme RM IM	First-order	74	28.75
	C10EE		n.d.	n.d.	12.05

**Table 3 molecules-28-02637-t003:** Specific productivities of New TAGs (P_spec_) obtained in each system, and respective biocatalyst costs.

	Lipozyme TL IM		Lipozyme RM IM		
Acyl Donor	P_spec_ (g New TAG/h.g Biocatalyst)	Biocatalyst Cost/kg New TAG (EUR)	P_spec_ (g New TAG/h.g Biocatalyst)	Biocatalyst Cost/kg New TAG (EUR)	Cost Ratio
C8:0	0.96	1.64	1.04	12.73	7.7
C10:0	1.65	0.95	1.79	7.36	7.7
C8EE	1.80	0.87	1.13	11.68	13.4
C10EE	1.62	0.97	1.87	7.06	7.3

**Table 4 molecules-28-02637-t004:** Flow rate (Q) and respective residence time (τ) estimated for each continuous reactions using Lipozyme TL IM or Lipozyme RM IM as a biocatalyst.

Parameters	Biocatalysts	
	Lipozyme TL IM	Lipozyme RM IM
Measured flow rate (Q, mL/min)	0.6	0.6
Enzyme bed volume (V, mL)	19.2	30.8
Void fraction (ε)	0.34	0.39
Residence time (τ, min)	10.9	20

## Data Availability

Data is available from the authors upon request.

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
