# Peer review of "Continuous Production of Dietetic Structured Lipids Using Crude Acidic Olive Pomace Oils"

_molecules, 2023, doi:10.3390/molecules28062637_

Round 1
Reviewer 1 Report
The paper shows the continuous production of dietetic structured lipids using acidic olive pomace oils. The research of the present issue has been the subject of study by some workers in the past, and, it is still an active area of research. Results looks in some sense positive, but its publication should be reconsidered after revision. I list the points which, in my opinion, should be improved.
1) Introduction section: The literature review on the lipase-catalyzed production of structured lipids needs to be substantially enhanced. The recently published articles are recommended to be included, for example: Food Chemistry, 2018,257:15-22;; Food Chemistry, 2017,227:397-403; Food Chemistry, 2016,194:1283-1292;.
2). The novelty of the work must be clearly addressed and discussed, compare your research with existing research findings and highlight novelty, (compare your work with existing research findings and highlight novelty),
3) The evaluation of the present research state, the further study trend can be given in the conclusion section
Author Response
Dear Reviewer,
In my name and in the name of all co-authors, we would like to thank you for the time and the helpful suggestions to improve our manuscript entitled: “Continuous Production of Dietetic Structured Lipids using Crude Acidic Olive Pomace Oils”.
Please find below the itemized rebuttal. We hope to have answered your comments accordingly. The changes in the manuscript are highlighted in yellow.
Best regards,
Suzana Ferreira-Dias
Comments and Suggestions for Authors
The paper shows the continuous production of dietetic structured lipids using acidic olive pomace oils. The research of the present issue has been the subject of study by some workers in the past, and, it is still an active area of research. Results looks in some sense positive, but its publication should be reconsidered after revision. I list the points which, in my opinion, should be improved.
Ans: We thank you for the favourable review of our manuscript.
1)Introduction section: The literature review on the lipase-catalyzed production of structured lipids needs to be substantially enhanced. The recently published articles are recommended to be included, for example: Food Chemistry, 2018, 257:15-22; Food Chemistry, 2017,227:397-403; Food Chemistry, 2016,194:1283-1292;
Ans:
Thank you very much for your recommendation to update the literature review in the introduction section. Please see the revised version of the Introduction section where the new sentences are highlighted in yellow.
As recommended, we included the three articles published by Xie and Zang in Food Chemistry, in 2016, 2017 and 2018.
2). The novelty of the work must be clearly addressed and discussed, compare your research with existing research findings and highlight novelty, (compare your work with existing research findings and highlight novelty),
Ans: We acknowledge the reviewer’s comment. In this revised version, we have rewritten the introduction section with the aim of clearly addressing the topic under investigation, while highlighting its original contribution regarding the current state-of-the-art knowledge.
3) The evaluation of the present research state, the further study trend can be given in the conclusion section
Ans: The conclusion section is an option of Molecules. However, we decided to include this section in the manuscript. Following your recommendation, the conclusion section was completed as follows:
”This is the first work on lipase-catalyzed continuous production of low-calorie SLs, in a packed-bed reactor, using crude acidic (12-29 % acidity) OPO in solvent-free media. Both immobilized lipases used (Lipozyme TL IM and Lipozyme RM IM) presented high activity either in acidolysis with caprylic or capric acid or in interesterification with their ethyl esters. The SLs were new TAGs where a medium-chain fatty acid was esterified at one of the positions sn-1 or sn-3 or in both (MLL or MLM molecules).
Yield of new TAGs around 50-60 % were observed for the interesterification reactions with C8EE or C10EE and acidolysis with capric acid, catalyzed by both biocatalysts. In the acidolysis with caprylic acid, new TAG yields were only around 30 %.
The biocatalysts maintained the activity along the 70-h continuous reactions except Lipozyme TL IM in presence of capric acid (half-life time of 228 h) and Lipozyme RM IM in the interesterification with C8EE (half-life time of 74 h). The acidity value of OPO used (12 to 29 % free fatty acids) did not affect the activity and stability of the biocatalysts.
Comparing specific productivities in new TAGs, similar values were obtained in presence of caprylic acid (0.96 vs. 1.04 g/h.g biocatalyst), capric acid (1.65 vs. 1.79 g/h. g biocatalyst) or C10EE (1.62 vs. 1.87 g/h.g biocatalyst), with Lipozyme TL IM or Lipozyme RM IM, respectively. Higher affinity for C8EE was observed for Lipozyme TL IM, when compared with Lipozyme RM IM, resulting in specific productivities of 1.80 and 1.13 g of new TAGs/h.g biocatalyst, respectively.Lipozyme TL IM, which is very prone to mechanical damage in batch stirred reactors, showed to maintain its activity along 70-h operation in a packed-bed reactor. Therefore, this biocatalyst is adequate for continuous processes. Moreover, due to its high productivity, stability and lower cost, Lipozyme TL IM demonstrated to be more promising than Lipozyme RM IM for MLM synthesis in continuous bioreactors.
Thus, using a crude oil that is a by-product of the olive oil industry, instead of refined oils currently used for SLs production, will contribute to decrease the process cost related to oil refining. In addition, the direct use of crude OPO increases process sustainability, reduces the environmental impact, promoting circular economy in the olive oil sector. Longer operation time in continuous bioreactors, as well as new bioreactor configurations, such as fluidized-bed reactors, together with the use of non-commercial biocatalysts, should be tested in further studies to decrease operation costs.”

Reviewer 2 Report
Authors describe the production of the so-called “dietetic structured lipids” from residual olive oil by acidolysis and interesterification with two commercial immobilized lipases.
This text is “more of the same” concerning the work of the authors, particularly the correspondence author, being more a review of their publications in last years. Indeed, 17 from the 36 references are self-citations!
Manuscript lacks in fundamental support of the assays, identification of the compounds obtained and justification of their advantages, interpretation of results, precision on calculations and concepts, accuracy in writing.
- The acronym MLM – lines 22, 66, 69, 97, … or MLL – line 199 – are not identified in full form all over the text. Also the concept of “novel TAG molecules” (line 199) or “new TAGs” (line 202 + 214) is not explained; which compounds were identified and in what proportion (because lipases are sn-1,3 regioselective and the new fatty acid can be incorporated in one or in both of the triglyceride positions)?
- Line 155 – 2.07% for palmitoleic acid – where is this value in Table 1?
- Line 158 – vital PUFA – concept of this?; … mainly esterified at position sn-2 in TAGs, …. ; this is not in accordance with line 68 and 69 where it is stated that LCFAs (it is supposed to be the oleic acid coming from the triolein) must be kept at position sn-2 (the so-called, but not described, MLM).
- Line 44 and 45 – the power 3 of 10 – must be written as kton or million tons, in accordance with line 51 for billion.
- Line 51 and 52 – uniformity in using the dot or the comma in numbers is missing.
- Line 121 – 125 (also 378 - 380) – to identify oxidation products by the absorbance reading at two wavelengths is not precise; several compounds absorb at the same wavelength!
- Line 136-138 (also 381 - 383) – to quantify chlorophyll pigments by absorbance is a very rough determination.
- Line 376 – the acidity of the samples was made by titration?
- Line 165-171 – what was the rationale for these choices? Why not others? Probably it comes from previous and similar works.
- Figure 1 and Figure 2 – the graphs A, B, C and D are not identified in the figures.
- The word “preferred” – line 223 and 227 (also line 333, 334) – is not very scientific, and it is used at direct observation of values without any interpretation.
- Legend of Figure 3 – continuous
- Line 310 – “type of bioreactor operation (batch or continuous)” , instead of “type of bioreactor (batch or continuous)”
- Line 327 (and Table 3) – a productivity cannot be defined in g/h – this is a production rate; a productivity must be defined per unity of reactor volume (that must also be defined because it is a packed-bed reactor). On the other hand, the specific productivity is well defined.
- Line 346 – the expected price to produce 1kg of structured lipids – what is 1 kg of structured lipids?
- Line 354 and 355 – it is a non-sense and a not supported sentence.
- Line 362 – “immobilized” is written twice.
- Line 360 – 363 – what are the enzymatic activity of both immobilized lipases used, and how were them determined?
- line 408, 409 – “the bioreactor operated for 70 h with a pause during the night” – I understand the logistic reason, but this affects the results, and this is not shown in graphs of Figures 1 and 2.
- Line 413 – Eq 3, instead of Eq 1
- Line 429 – 10 g of each immobilized lipase – it is a practical information to operate the reactors, but the real information is the equivalent in enzymatic activity, not shown.
- Line 431 – “the void fraction and residence time were calculated according to Xu et al [13]” - these are common parameters; it is non-sense to refer a reference for them.
- Line 432 and 433 – why the residence times were quite different (11 and 20 min) if the flow rate was the same and the void fraction was very similar (0.34 and 0.39)?
- Line 473 – the (instead of “de”)
- line 483 – 492 – conclusions are very poor. No real conclusions at all. What are the structured lipids (or MLM synthesis!) produced? Why are they “high value added”? To which applications?
Author Response
Dear Reviewer,
In my name and in the name of all co-authors, we would like to thank you for the time and the helpful suggestions to improve our manuscript entitled: “Continuous Production of Dietetic Structured Lipids using Crude Acidic Olive Pomace Oils”.
Please find below the itemized rebuttal. We hope to have answered your comments accordingly. The changes in the manuscript are highlighted in yellow.
Best regards,
Suzana Ferreira-Dias
Comments and Suggestions for Authors
Authors describe the production of the so-called “dietetic structured lipids” from residual olive oil by acidolysis and interesterification with two commercial immobilized lipases.
This text is “more of the same” concerning the work of the authors, particularly the correspondence author, being more a review of their publications in last years. Indeed, 17 from the 36 references are self-citations!
Ans: We tried not to have a very large introduction and present this work as a follow up of our previous studies on Low-calorie SLs production (not on the other types of SLs we have been working on, such as trans-free fat blends for margarine industry or human milk fat substitutes). As requested, we complemented the introduction with other studies. As far as we know, we have been the ones working with crude olive pomace oils for SLs production. Most of the studies have been carried out with refined oils. The use of crude high acidic oils and continuous bioreactors have advantages in terms of process costs and industrial scale-up, as referred in this manuscript.
Manuscript lacks in fundamental support of the assays, identification of the compounds obtained and justification of their advantages, interpretation of results, precision on calculations and concepts, accuracy in writing.
Ans: To our opinion, the full description of the methods previously reported in the literature is not needed. Moreover, since we have been working in the field of SLs for several years, and we have a considerable number of publications in this field, we have the risk of self-plagiarism because it is not possible to describe the methods currently used in very different ways.
In accordance with the Reviewer recommendation and to facilitate the understanding of our results, we include in this version of the manuscript an example of a chromatogram of the initial blend and of the interesterified blend.
Concerning the identification of the compounds, as said in the first version of the manuscript, “Each peak in the chromatograms was identified by comparing it to pure compound standards and with the chromatograms presented in the European Standard EN 14105 [36].”
The concepts of yield and conversion are those currently used in biochemical engineering. In order to clarify these concepts and to explain how the calculations were performed, the following initial sentence, “The quantification was determined by the yield of novel produced TAG molecules (ratio between the amount of new TAG and the initial TAG), as well as TAG and FFA or ethyl esters conversion (ratio between the amount of ethyl ester, FFA or TAG consumed and the corresponding initial quantities).”
was rewritten as follows:
“In each chromatogram, the area of each peak was divided by the area of the peak of the internal standard (IS). The amounts of initial TAGs were proportional to the sum of the areas of all TAG peaks (Ainitial TAG) divided by the area of the peak of the IS (AIS), in the initial fat blend. Along bioreactor operation, the same procedure was used to quantify the new TAGs that appear as new peaks in TAG region of the chromatogram (Anew TAG) (Fig. 4). Therefore, New TAG yield (Y, %) was calculated as follows (eq. 4), where the numerator corresponds to the peaks of the chromatogram at time t, and the denominator corresponds to the peaks of the chromatogram of the initial blend:
EQUATION IN THE WORD FILE (4)
TAG, FFA or ethyl esters conversions were calculated as the ratio between the amount of TAGs (ethyl ester or FFA) consumed and the corresponding initial quantities (in terms of peak areas divided by the area of the IS peak, as explained for New TAG yield)..
- The acronym MLM – lines 22, 66, 69, 97, … or MLL – line 199 – are not identified in full form all over the text. Also, the concept of novel TAG molecules” (line 199) or “new TAGs” (line 202 + 214) is not explained; which compounds were identified and in what proportion (because lipases are sn-1,3 regioselective and the new fatty acid can be incorporated in one or in both of the triglyceride positions)?
Ans: The definition of MLM-type SLs was not presented in the abstract due to the maximum of 200 words allowed and because the acronym MLM is well known by those who work in structured lipids. We understand that, since Molecules is not a journal for experts in lipids, the definition of MLM is missing here. Thus, the abstract was modified as follows (209 words):
“Crude olive pomace oil (OPO) is a by-product of olive oil extraction. In this study, low-calorie structured Triacylglycerols (TAGs) were produced by acidolysis of crude OPO with medium-chain fatty acids (caprylic, C8:0; capric, C10:0) or interesterification with their ethyl esters forms (C8EE, C10EE). These new TAGs present long-chain fatty acids (L) at position sn-2 and medium-chain fatty acids (M) at positions sn-1,3 (MLM). Crude OPO exhibited high acidity (12.05-28.75% free fatty acids), and high contents of chlorophylls and oxidation products. Reactions were carried out continuously in a packed-bed bioreactor for 70 h, using sn-1,3 regioselective commercial immobilized lipases (Thermomyces lanuginosa lipase, Lipozyme TL IM; and Rhizomucor miehei lipase, Lipozyme RM IM), in solvent-free media, at 40°C. Lipozyme RM IM presented higher affinity for C10:0 and C10EE. Lipozyme TL IM preferred C10:0 over C8:0 but C8EE over C10EE. Both biocatalysts showed high activity and operational stability and were not affected by OPO acidity. New TAGs yield ranged 50-60 % and the productivity ranged 9.56-18.67 g NewTAG/h. Lipozyme RM IM cost is more than 8-fold Lipozyme TL IM cost. Therefore, using Lipozyme TL IM and crude acidic OPO in continuous bioreactor, will contribute for process sustainability for structured lipid production by lowering the cost of the biocatalyst and avoiding oil refining.”
The definition of MLM was presented in the Introduction section in lines 66-69 (now lines 95-104). However, to clarify this sentence, medium-chain fatty acids (MCFAs) and long-chain fatty acids (LCFAs) were replaced by “M” and “L”, respectively, and additional information was added, as follows:
Lines 95-104: “Among SLs, dietetic structured lipids, also known as low-calorie TAGs, are a type of lipids containing a specific combination of medium-chain fatty acids (M) at positions sn-1 and sn-3, and a long-chain fatty acid (L) at position sn-2 (MLM). In vegetable oils, mono- and polyunsaturated long-chain fatty acids, namely essential FAs, are preferentially located at position sn-2. Medium-chain fatty acids have lower caloric value than long-chain FAs and are metabolized in the liver like glucose, and not accumulated as fat in the human body. Therefore, MLM SLs will provide the essential FAs for the humans and, at the same time, will contribute for obesity control due to their lower caloric value than that of natural fats (c.a 5 kcal/g against 9 kcal/g natural fats [11, 12]. These properties make MLM useful for reducing the possibility of obesity and promoting proper body function.”
Concept of novel TAG molecules” (line 199) or “new TAGs” (line 202 + 214) is not explained;
Ans: Sorry but we do not understand your question: novel TAG molecules or new TAGs are the same. As previously explained in lines 202-205, during interesterification, if only one fatty acid substitution occurs at position sn-1 or sn-3, the new TAG molecule will be of the type MLL (equivalent to LLM); if the substitution occurs at both positions, the new TAG will be of MLM type. Therefore, new TAGs can be MLL or MLM. Due to acyl migration, the formation of TAGs containing caprylic or capric acid at position sn-2 (MML, LML or MMM) may also occur but in lesser extent. Due to the lack of standards for this type of TAGs with mixed FA in specific positions, we could not identify the new TAGs formed. We could only identify the groups of compounds by comparison with standard chromatograms. The information was completed as follows:
Lines 265-272: “The novel TAG molecules generated might be of MLL or MLM type, depending on the FA substitution achieved with the acyl donor involved (one or two substitutions of a long-chain FA by caprylic or capric acid, at positions sn-1,3, respectively) [8]. Due to acyl migration, the formation of TAGs containing caprylic or capric acid at position sn-2 (MML, LML or MMM) may also occur but in lesser extent [20]. Since these reactions create intermediate molecules (monoacylglycerols, MAGs, and diacylglycerols, DAGs), the conversion values of TAGs are always higher than the yield of new TAGs along the reactions.”
- Line 155 – 2.07% for palmitoleic acid – where is this value in Table 1?
Ans: You are right. There is a mistake. It was corrected as follows: palmitoleic acid (C16:1) contributed for 1.32-1.46 %
- Line 158 – vital PUFA – concept of this?… mainly esterified at position sn-2 in TAGs, …. ; this is not in accordance with line 68 and 69 where it is stated that LCFAs (it is supposed to be the oleic acid coming from the triolein) must be kept at position sn-2 (the so-called, but not described, MLM).
Ans: You are right: the sentence in lines 68 and 69 was not clear. It was modified as already referred.
In line 158, “vital PUFA” was replaced by “oleic and linoleic acids”. These acids are mainly esterified at position sn-2 in TAG molecules of olive pomace oil (similar to olive oil)
- Line 44 and 45 – the power 3 of 10 – must be written as kton or million tons, in accordance with line 51 for billion.
Ans: We are grateful for this correction, and we have changed as requested.
- Line 51 and 52 – uniformity in using the dot or the comma in numbers is missing.
Ans: We are grateful for this correction, and we have changed as requested.
- Line 121 – 125 (also 378 - 380) – to identify oxidation products by the absorbance reading at two wavelengths is not precise; several compounds absorb at the same wavelength!
Ans: You are right. However, the aim of this study was not the identification of oxidation products. For olive oil and olive pomace oil, according to the European Commission regulation, K232 and K270 are used as quality parameters related with the oxidation level [39]. We used these parameters for comparison and, when appropriate to follow the oxidation of olive oil along storage. The reference to the legislation [39] that was in the Materials and Methods section, was moved to the section “2.1. Crude olive pomace oil characterization”
- Line 136-138 (also 381 - 383) – to quantify chlorophyll pigments by absorbance is a very rough determination.
Ans: You are right. However, as for oxidation products, the aim of this study was not the identification of chlorophyll pigments. Using the referred spectrophotometric method, we could compare the three olive pomace samples in terms of green pigment content. All the samples came from the same olive pomace oil extraction plant, which receives olive pomace from the same olive mills along the olive oil extraction period. As you can see, the olive pomace samples were not very much different in terms of green pigments content.
- Line 376 – the acidity of the samples was made by titration?
Ans: Yes, according to the methodology indicated in the European Commission regulation, as referred in Materials and Methods section.
- Line 165-171 – what was the rationale for these choices? Why not others? Probably it comes from previous and similar works.
Ans: As referred in the introduction section (lines 163-167), previous studies were performed by our group to valorize crude OPO as feedstock for the production of SLs, in batch bioreactors.
- Figure 1 and Figure 2 – the graphs A, B, C and D are not identified in the figures.
Ans: We are grateful for this correction, and we have changed as requested.
- The word “preferred” – line 223 and 227 (also line 333, 334) – is not very scientific, and it is used at direct observation of values without any interpretation.
Ans: we replaced “preferred” in lines 223 and 227 (now lines 323, 325 and 327) by “showed higher affinity for…”. It is known that lipases presented different behaviors (affinities /preferences) for different molecules.
The following interpretation was added (lines 297-317):
“The observed behavior for both biocatalysts is characteristic of lipases which present higher affinity for longer-chain fatty acids than for medium- or short-chain fatty acids, conversely to esterases [43]. Therefore, a higher activity is expected with increasing fatty acid chain-length.
The observed results can be also explained by the concept of Log P (Hansch parameter) applied to the acyl donors used in each reaction system. Laane et al. [44] proposed for the first time, the use of Log P to assess the biocompatibility of organic solvents: solvents with Log P lower than 2 are harmful for the biocatalysts because they are very hydrophilic and remove the essential water layer of the biocatalysts, between 2 and 4, their behavior is unpredictable, and values higher than 4 indicate that no inactivation of the biocatalyst will occur. From data published by PubChem (https://pubchem.ncbi.nlm.nih.gov/), the Log P values are as follows: Log P C8:0 = 3.05; Log P C10:0 = 4.09; Log P C8EE = 3.842; and Log P C10EE = 4.861.
Both capric acid and ethyl decanoate have Log P values higher than 4. When Lipozyme RM IM was used, the highest productivities in SLs were obtained in presence of these acyl donors. Lipozyme TL IM seemed not to be so much affected with more hydrophilic molecules since it showed similar results also in presence of C8EE, with a Log P value lower than 4.0. Either for Lipozyme TL IM or for Lipozyme RM IM, the lowest productivities in SLs were obtained in presence of caprylic acid, which has a Log P value lower than 4. Thus, a possible inactivation of the biocatalysts caused by the more hydrophilic acyl donors, may explain the observed results.”
- Legend of Figure 3 – continuous
Ans: Corrected
- Line 310 – “type of bioreactor operation (batch or continuous)” , instead of “type of bioreactor (batch or continuous)”
Ans: Corrected
- Line 327 (and Table 3) – a productivity cannot be defined in g/h – this is a production rate; a productivity must be defined per unity of reactor volume (that must also be defined because it is a packed-bed reactor). On the other hand, the specific productivity is well defined.
Ans: According to Doran (1995) (Reference 42 in the manuscript), the production rate is also called productivity. We decided to delete the columns with production rates and keep only the one with the specific productivity. This information is sufficient for comparison purposes between the two lipases and the different systems.
- Line 346 – the expected price to produce 1kg of structured lipids – what is 1 kg of structured lipids?
Ans: We considered as structured lipids, the new TAGs produced. Thus, considering a flow rate of 0.6 mL/min, after 70 h running, we have 2520 mL of reaction medium. If, for example, in the system C10EE + RM IM, we have 56.4 % new TAG yield, it means that a volume of 1421 mL of new TAGs was obtained. Considering the density of fats (0.92 g/mL), it means that we have 1.37 kg of new TAGs, i.e. of structured lipids.
- Line 354 and 355 – it is a non-sense and a not supported sentence.
Ans: the sentence was deleted.
- Line 362 – “immobilized” is written twice.
Ans: Corrected
- Line 360 – 363 – what are the enzymatic activity of both immobilized lipases used, and how were them determined?
Ans: As you know, there is no relationship between the hydrolytic activity of a lipase and its interesterification activity. Please see our work:
Nunes, P.A., Pires-Cabral, P., Guillén, M., Valero, F., Ferreira-Dias, S. (2012) Optimized production of MLM triacylglycerols catalyzed by immobilized heterologous Rhizopus oryzae lipase, Journal of the American Oil Chemists’ Society, 89: 1287–1295 (http://dx.doi.org/10.1007/s11746-012-2027-9).
The interesterification activity of both biocatalysts supplied by the manufacturer was added: Lipozyme TL IM, 250 IUN/g; Lipozyme RM IM: 275 IUN/g. This information was added to the manuscript (lines 473 and 474).
Since they have similar activities, we decided to use the same amount of biocatalyst (10 g) in the bioreactor and compare the results in terms of New TAGs formed along 70 h continuous operation.
- line 408, 409 – “the bioreactor operated for 70 h with a pause during the night” – I understand the logistic reason, but this affects the results, and this is not shown in graphs of Figures 1 and 2.
Ans: Before turning off the bioreactor, a sample was taken and analyzed. The next day, after waiting three residence times to attain the steady state, another sample was taken. No differences were detected between both samples.
- Line 413 – Eq 3, instead of Eq 1
Ans: Corrected
- Line 429 – 10 g of each immobilized lipase – it is a practical information to operate the reactors, but the real information is the equivalent in enzymatic activity, not shown.
Ans: please see the answer about enzyme activity.
- Line 431 – “the void fraction and residence time were calculated according to Xu et al [13]” these are common parameters; it is non-sense to refer a reference for them.
Ans: we could have estimated the porosity by methods such as porosimetry but we estimated the void fraction by volume displacement, according to the methodology followed by Xu et al. [27]. That is the reason why we put the reference.
- Line 432 and 433 – why the residence times were quite different (11 and 20 min) if the flow rate was the same and the void fraction was very similar (0.34 and 0.39)?
Ans: This difference may be explained by differences in column packaging originating different bed volumes. Xu et al. [27] obtained different values of void fraction in a packed-bed column filled twice with Lipozyme RM IM (0.44 ± 0.01 and 0.47 ± 0.02). For Lipozyme TL IM in a packed-bed reactor, a void fraction of 0.51 ± 0.1 was estimated by Xu et al [28].
This information was added to the manuscript (lines560-564).
- Line 473 – the (instead of “de”)
Ans: Corrected
- line 483 – 492 – conclusions are very poor. No real conclusions at all. What are the structured lipids (or MLM synthesis!) produced? Why are they “high value added”? To which applications?
Ans: The conclusion section is an option of Molecules. However, we decided to include this section in the manuscript. The applications of low-calorie SLs, as well as examples of some of these products already in the market, were added to the Introduction section, since they were not conclusions of our study.
Following your recommendation, the conclusion section was completed as follows:
”This is the first work on lipase-catalyzed continuous production of low-calorie SLs, in a packed-bed reactor, using crude acidic (12-29 % acidity) OPO in solvent-free media. Both immobilized lipases used (Lipozyme TL IM and Lipozyme RM IM) presented high activity either in acidolysis with caprylic or capric acid or in interesterification with their ethyl esters. The SLs were new TAGs where a medium-chain fatty acid was esterified at one of the positions sn-1 or sn-3 or in both (MLL or MLM molecules).
Yield of new TAGs around 50-60 % were observed for the interesterification reactions with C8EE or C10EE and acidolysis with capric acid, catalyzed by both biocatalysts. In the acidolysis with caprylic acid, new TAG yields were only around 30 %.
The biocatalysts maintained the activity along the 70-h continuous reactions except Lipozyme TL IM in presence of capric acid (half-life time of 228 h) and Lipozyme RM IM in the interesterification with C8EE (half-life time of 74 h). The acidity value of OPO used (12 to 29 % free fatty acids) did not affect the activity and stability of the biocatalysts.
Comparing specific productivities in new TAGs, similar values were obtained in presence of caprylic acid (0.96 vs. 1.04 g/h.g biocatalyst), capric acid (1.65 vs. 1.79 g/h. g biocatalyst) or C10EE (1.62 vs. 1.87 g/h.g biocatalyst), with Lipozyme TL IM or Lipozyme RM IM, respectively. Higher affinity for C8EE was observed for Lipozyme TL IM, when compared with Lipozyme RM IM, resulting in specific productivities of 1.80 and 1.13 g of new TAGs/h.g biocatalyst, respectively.Lipozyme TL IM, which is very prone to mechanical damage in batch stirred reactors, showed to maintain its activity along 70-h operation in a packed-bed reactor. Therefore, this biocatalyst is adequate for continuous processes. Moreover, due to its high productivity, stability and lower cost, Lipozyme TL IM demonstrated to be more promising than Lipozyme RM IM for MLM synthesis in continuous bioreactors.
Thus, using a crude oil that is a by-product of the olive oil industry, instead of refined oils currently used for SLs production, will contribute to decrease the process cost related to oil refining. In addition, the direct use of crude OPO increases process sustainability, reduces the environmental impact, promoting circular economy in the olive oil sector. Longer operation time in continuous bioreactors, as well as new bioreactor configurations, such as fluidized-bed reactors, together with the use of non-commercial biocatalysts, should be tested in further studies to decrease operation costs.”

Round 2
Reviewer 1 Report
The manuscript has been improved according with my suggestions. Thus my recommendation is to publish the paper.
Reviewer 2 Report
The manuscript was quite improved in this revised version and can now be published.
A last advice is:
In the legend of the new Figure 4, in line 4, …-IS – peak of the … (instead of pak)